# Design of a 35 kW Solar Cooling Demonstration Facility for a Hotel in Spain

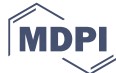

**Pedro J. Martínez [1],\***  **, Pedro Martínez [1], Victor M. Soto [2]  , Luis A. Bujedo [3] and Juan Rodriguez [4]**

1   Departamento de Ingeniería Mecánica y Energía, Universidad Miguel Hernández, Avda. de la Universidad, s/n, 03202 Elche, Spain; pedro.martinez@umh.es
2   Departamento de Termodinámica Aplicada, Universitat Politècnica de Valencia, 46022 Valencia, Spain; vsoto@ter.upv.es
3   CARTIF Centro Tecnológico, 47151 Boecillo, Spain; luibuj@cartif.es
4   Fraunhofer Institut für Energiewirtschaft und Energiesystemtechnik, 34119 Kassel, Germany; juan.rodriguez.santiago@iee.fraunhofer.de
\*   Correspondence: pjuan.martinez@umh.es

**Abstract:** Solar cooling systems have the advantage of the coincidence between the hours of cooling demand and the hours of solar radiation availability, and they can contribute to reduce the energy consumption in buildings. However, the high cost of thermal solar cooling facilities with absorption chillers, maintenance issues, legionella risk and water consumption (associated to the necessary cooling tower) have limited the use of these systems to demonstration projects. A simplified Transient System Simulation Tool (TRNSYS) model was developed to provide the owner of the demonstration facility the information he needs for design decision-making. This model was validated with experimental data registered in a solar cooling system designed and built by the authors. Different collector field surfaces, hot water storage tank volumes, and absorption machine driving temperatures were analyzed for a hotel demonstration facility. In terms of the energy delivered to the absorption chiller the optimum dimensioning corresponded to the lowest values of the driving temperature (75 °C) and specific storage volume (15 Lm$^2$). From an economic point of view, the saving of 1515 euros per year when compared with an electric compression chiller does not compensate the investment of 3000 euros per kW of cooling capacity that cost the thermal solar cooling facility.

**Keywords:** solar cooling; single-effect absorption chiller; TRNSYS; demonstration facility

## 1. Introduction

It is estimated that, in the coming years, the consumption of electricity for cooling in buildings will grow, which is attributable to several factors, such as: the increase in ambient temperature, the growth of comfort expectations, the perception of increased productivity linked to thermal comfort, and increased internal load due to electronic equipment. This increase in energy consumption due to refrigeration systems will result also in a significant increase in $CO_2$ emissions.

The aim of Article 9 of the 2010 Energy Performance of Buildings Directive (EPBD) [1] is that all new buildings have nearly zero energy needs by the end of 2020. It also establishes that the low amount of energy required by buildings should be supplied to a significant extent from renewable sources.

Solar cooling systems agree with this idea since it makes sense cooling the buildings with the same energy that causes much of the conditioned spaces cooling load. This solution is especially suitable for commercial buildings where there is a coincidence between the hours of cooling demand and the hours of solar radiation availability.

Solar cooling will account for 17% of world air conditioning by 2050 according to the International Energy Agency [2]. However, the initial cost of thermal solar cooling systems with absorption machine is very high with payback times higher than their lifetime [3]. Nowadays, from an economic point of view, the vapor compression systems driven by photovoltaic solar collectors are the most attractive solar air conditioning systems [4]. For residential applications in Central Spain, Infante Ferreira and Kim calculated an investment cost of 1050–1550 euros per kW of cooling capacity for the PV and vapor compression system and 2675 euros per kW of cooling capacity for the thermal solar cooling system with flat-plate solar collector and single-effect absorption machine.

In addition to their economic feasibility, photovoltaic (PV) solar collectors combined with electrical chillers with a high energy efficiency ratio (EER) need smaller heat rejection systems. This avoids the risk of legionella associated to the use of cooling towers with LiBr-$H_2O$ absorption chillers and the corresponding water consumption [5]. These are the main reasons why most thermal solar cooling facilities currently in operation are part of demonstration projects.

The performance of thermal solar air conditioning systems is usually evaluated through detailed simulations, which require specific software to establish the system's model. Possibly the most well-known simulation tool for solar thermal systems is Transient System Simulation Tool (TRNSYS) [6], which is a widely used modular simulation program. Solar cooling facilities are complex systems that are better designed with the help of detailed simulations [7].

The model of the LiBr-$H_2O$ single-effect absorption chiller is important to simulate a thermal solar air-conditioning system. TRNSYS includes an absorption chiller model in its library that needs a specific input file based on performance data provided by the manufacturer. This fact limits its use when those data are not available [8].

The design of a solar air conditioning facility basically involves the design of the solar collector field and the storage tank [9]. Detailed models have been developed in the environment of TRNSYS and used for the analysis of the effect of the main design variables on the performance of the thermal solar cooling facility [10]. These models sometimes include the control system [11].

Simplified methods have also been developed to evaluate the long-term performance of solar cooling systems in terms of the solar fraction. Klein and Beckman [12] developed the φ, f-chart method for closed-loop systems with finite storage where the load (thermally-driven cooling machine) is characterized by a minimum useful temperature. Oliveira [13] presented a method based on the calculation of two utilizability values related to the temperature at which the hot water is delivered to the absorption machine and the temperature at which hot water returns from the chiller.

Using TRNSYS, a simple model can also be developed that offers the owner of the demonstration facility the information he needs to decide the dimensions of the solar collector field and the hot water storage tank. This design tool would not include the detailed models of the absorption chiller, the cooling tower or even the cooling load (building).

The hypothesis of not modeling the building cooling load is valid if it is assumed that in the demonstration solar cooling facility the absorption machine participates in the building's base cooling demand and whenever it is in operation it will perform at full load.

The objective of this work was to develop a simplified TRNSYS design model for small capacity solar cooling demonstration facilities. This model allows studying the effect of critical design parameters on the performance of this type of facilities, and establishes the basis for decision-making by the hotel owner.

## 2. Materials and Methods

The objective of this work was the design of the components of a solar air conditioning facility for the Foxá 3 Cantos hotel located in Madrid. The hotel owner wanted to take advantage of an existing field of flat-plate solar collectors to carry out a demonstration project of solar cooling. The flat-plate solar collector (Vitosol 100), its orientation (south) and inclination (60°) are starting data, as well as the

cooling capacity of the absorption machine Yazaki WFC SC10 (35 kW), which accounts for only 3% of the installed cooling capacity.

As mentioned above, the design of a solar air conditioning facility basically involves the design of the solar collector field and the storage unit. That is why this study has been aimed at evaluating the fraction of incident solar radiation that is capable of being used by the absorption chiller for the production of cold water at 7 °C. To this end, the effect of the following parameters on the performance of the system was evaluated: solar collector field surface, hot water storage volume, and heat storage temperature required to drive the absorption machine when it is idle. The inclusion of the drive temperature analysis is considered essential since it determines both the efficiency of the solar collector field and the EER of the absorption chiller.

The available solar radiation and the rest of the climatic variables of the locality in which the installation is located have been obtained through the METEONORM program [14]. Figure 1 shows an outline of the model developed in the TRNSYS program [6] for the design of the facility.

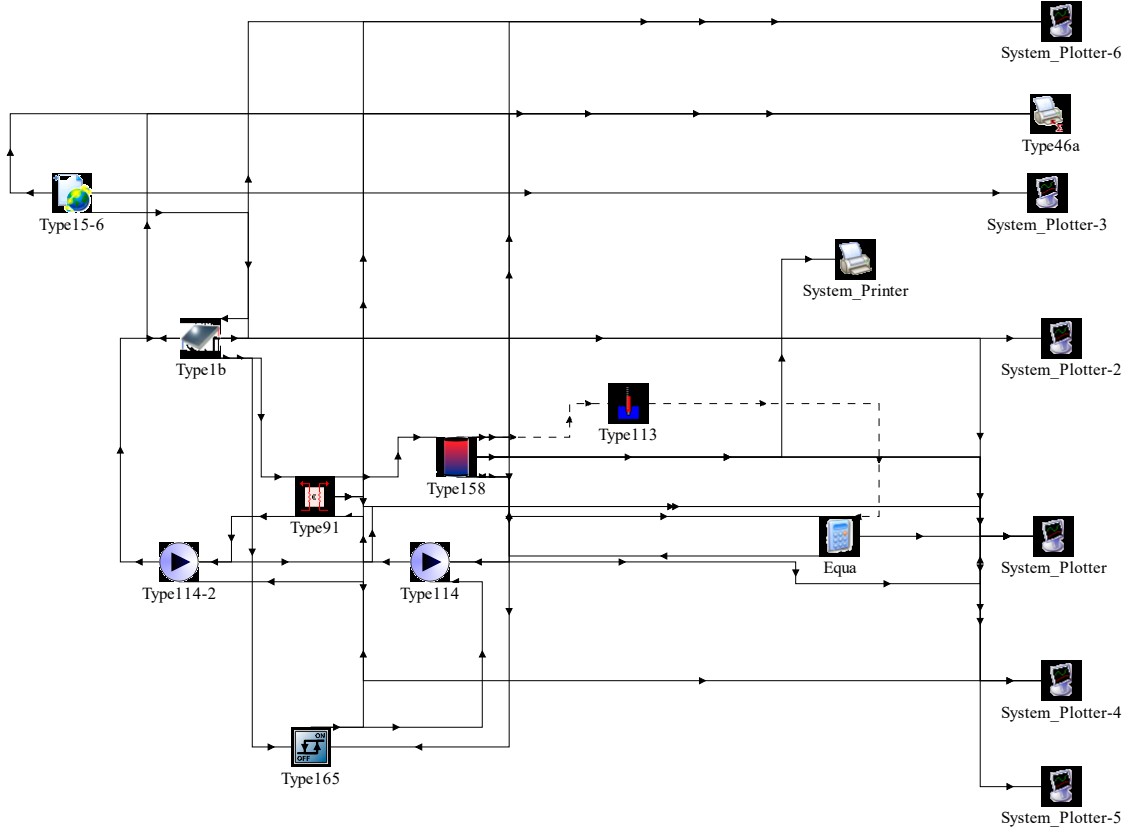

**Figure 1.** Scheme of the system model built in Transient System Simulation Tool (TRNSYS).

Its operating principle is as follows. A solar collector field (whose surface is one of the study variables) transfers the collected energy to a storage tank (whose volume is another of the study variables). When the water temperature in the storage tank reaches a certain value (set-point: operating temperature plus 2.5 °C), a power of 50 kW is extracted from it, which is the nominal operating power of the absorption chiller. The energy extraction stops when the temperature of the tank decreases below the set-point temperature minus 2.5 °C.

The TRNSYS model integrates the different energy flows. In this case, the energy extracted from the storage tank has been calculated for each combination of the design parameters. This will allow us to express the potential of each combination as the fraction of the solar radiation incident in the collector field that can be used for driving the absorption chiller.

Type 1 of TRNSYS models the efficiency of the flat-plate solar collector using a quadratic equation. The coefficients provided to the model were those corresponding to the collector model installed in the hotel (Vitosol-100): $a_0 = 0.84$; $a_1 = 3.36$ (W/m$^2$K) and $a_2 = 0.013$ (W/m$^2$K$^2$). This efficiency depends on the temperature of the fluid circulating through the flat-plate collector (T) according to Equation (1).

$$\eta_{col} = a_0 - a_1 \frac{(T - T_{amb})}{I} - a_2 \frac{(T - T_{amb})^2}{I} \tag{1}$$

Type 158 was used for the heat storage tank. This model simulates the operation of a cylindrical tank that has been assigned a loss coefficient of 0.77 W/m$^2$ °C. The tank has been divided into a total of 10 nodes in order to adequately model their degree of stratification. The tank interacts with two water flows, one that connects it to the heat exchanger (where the energy from the solar collector field comes from) and the other that connects it to the load (absorption machine) and through which the nominal drive power (50 kW) is extracted when the temperature of the water in the tank is higher than the set-point temperature.

As for the control system, a differential thermostat (Type 165) is used to drive the pumps (Type 114) as a function of the temperature difference between the collector outlet and the lower part of the storage tank. Finally, a thermostat (Type 113) is used to detect when the storage tank reaches the required driving temperature.

This work focuses on the design of the solar collector field and the hot water storage tank which are the main design variables of a solar cooling installation, as well as on the driving temperature of the absorption chiller as a control parameter. It has been chosen to simplify the model in relation to the rest of the components using the operating curves of the absorption chiller provided by the manufacturer in which only the driving temperature is varied. Both cooling water inlet temperature and chilled water outlet temperature are considered constant and equal to their nominal values.

### 2.1. Design Variables Levels

The energy delivered from the hot water storage tank to the absorption chiller for the production of cold water was evaluated, as a function of the collector field surface, volume of the hot water storage tank and driving temperature of the absorption machine.

The levels selected for the driving temperature were 75, 80, and 85 °C. A hysteresis band in the thermostat of 5 °C was considered for all three temperatures in order to guarantee the difference in the thermal level of the absorption chiller. That is, in the first case (set-point temperature = 75 °C), when 77.5 °C is reached in the storage tank, hot water is sent to the chiller until the temperature of the tank drops to 72.5 °C, which is a value of 2.5 °C higher than the minimum operating temperature set by the manufacturer of the absorption machine. In the second case, the operation of the machine takes place with water between 82.5 and 77.5 °C and in the third between 87.5 and 82.5 °C. The temperature of 87.5 °C is only 0.5 °C lower than the chiller nominal driving temperature (88 °C).

The levels selected for the specific collector area were: 2.5 m$^2$/kW$_{cool}$, 3 m$^2$/kW$_{cool}$, 3.5 m$^2$/kW$_{cool}$, and 4m$^2$/kW$_{cool}$. The specific collector area is defined as the ratio between the surface of the collector field and the nominal power of the absorption chiller (35 kW). In the present study, the specific surfaces analyzed correspond to collector surfaces of 87.5 m$^2$ (35 collectors), 105 m$^2$ (42 collectors), 122.5 m$^2$ (49 collectors), and 140 m$^2$ (56 collectors), respectively. The values tested are common in the design of this type of facilities [15].

The specific storage volumes selected were: 15 L/m$^2$, 32.5 L/m$^2$ and 50 L/m$^2$. The volume of the hot water storage tank (chiller driving medium) is specified per unit of solar collector surface [16]. The volume of the storage tank will therefore vary from 1.3 m$^3$ corresponding to the collector surface and specific storage values of 87.5 m$^2$ and 15 L/m$^2$, to 7 m$^3$ corresponding to the collector surface and specific storage values of 140 m$^2$ and 50 L/m$^2$.

*2.2. Validation of the TRNSYS Model*

The model developed in the TRNSYS program was validated with the data from a solar cooling facility designed by the authors and built in the Miguel Hernández University of Elche [17] (Figure 2). This facility conditions: 200 m² of laboratories and offices. It has a collector field of 38.4 m² composed of 16 flat-plate collectors oriented toward the south with a 30 degrees slope. The hot water storage tank has a volume of 1 m³.

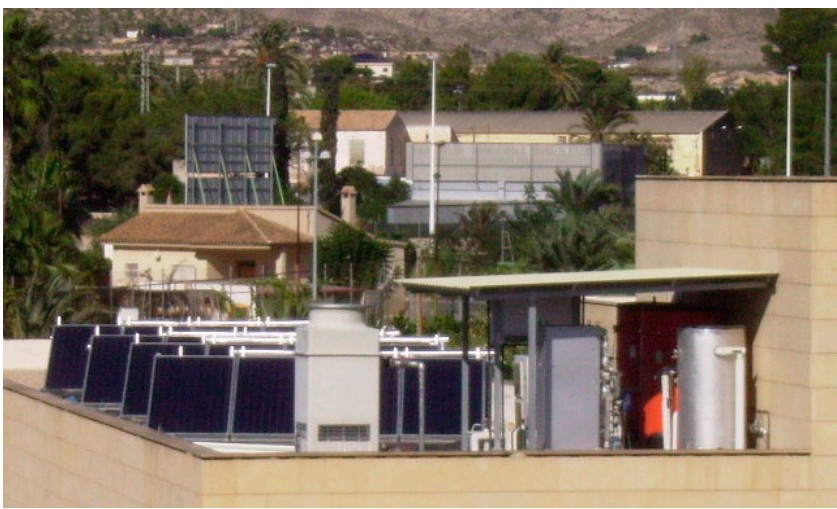

**Figure 2.** View of the solar cooling system at Miguel Hernández University of Elche.

Figure 3 shows a scheme of the solar cooling system. The solar collectors supply hot water to the storage tank to drive the absorption chiller. If the solar radiation is not enough, a backup vapour compression chiller covers the cooling demand. The cold water supplied to the fan coils comes either from the solar facility or the backup system, depending on the temperature of the cold water storage by actuating the three-way valves V1 and V2 [10].

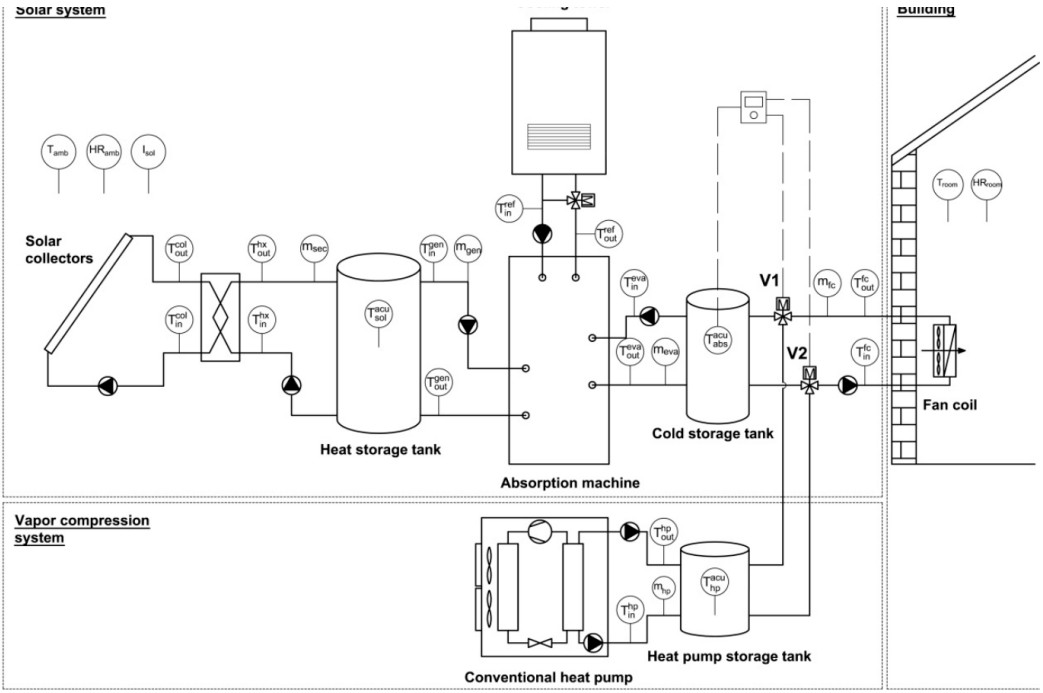

**Figure 3.** Scheme of the solar cooling system.

Figure 3 also shows the positions of the sensors in the system. Temperatures were registered by Pt-100 temperature sensors and water flow rates by electromagnetic flow meters. A first-class WMO pyranometer was used to measure solar irradiance incident on the collectors. Outdoor and indoor dry-bulb temperature and humidity ratio were also registered as well as water and electricity consumptions.

Table 1 shows the difference between the results obtained with the TRNSYS model proposed in this work (Figure 1) and those registered in the installation and published in [17], in terms of the fraction of the solar radiation used to drive the absorption machine.

**Table 1.** Registered and simulated fractions of the incident solar radiation used to drive the absorption chiller for driving temperatures of 75 and 80 °C.

| Driving Temperature (°C) | Useful Fraction Registered | Useful Fraction TRNSYS |
|:---:|:---:|:---:|
| 75 | 0.348 | 0.4 |
| 80 | 0.293 | 0.365 |

To obtain the TRNSYS results of Table 1 and validate the model, the input data were those of the Elche facility, that is, the typical meteorological year, the surface (38.4 m$^2$), the efficiency curve ($a_0 = 0.818$; $a_1 = 3.47$ (W/m$^2$K) and $a_2 = 0.0101$ (W/m$^2$K$^2$)) and the inclination (30°) of the solar collectors and the storage volume (1 m$^3$). Once the model for the Elche facility has been validated, it is applied for the design analysis of the demonstration facility in Madrid.

The differences of 0.052 (75 °C) and 0.072 (80 °C) are due to several reasons: the use of a typical meteorological year, the difference between the performance of the solar collectors with respect to the efficiency curve supplied by the manufacturer, and thermal losses in pipes. Although meaningful, the relevance of the obtained differences will be evaluated at the end of this study.

To characterize the climatic conditions in summer (June–September), a typical meteorological year was obtained using METEONORM (2007) for Madrid and Elche [14]. Table 2 shows the average daily radiation received per square meter of inclined collector surface for the months that make up the summer season (June-September) in Madrid and Elche.

**Table 2.** Average daily radiation received per square meter of inclined collector surface.

| Collectors Slope (°) | Energy (kWh/m$^2$-day) Elche | Energy (kWh/m$^2$-day) Madrid |
|:---:|:---:|:---:|
| 10 | 7.19 | 6.90 |
| 20 | 7.24 | 6.96 |
| 30 | 7.10 | 6.86 |
| 40 | 6.80 | 6.59 |
| 60 | 5.72 | 5.60 |

It can be seen that the ideal slope of the solar collectors for this application is 20°, although for ease of construction of the supports, in these cases an inclination of 30° is normally adopted (sin 30° = 0.5). This is a value that could be considered if the owner decided to build a specific collector field for the absorption machine. The present study was carried out for an inclination of 60°, which is the current slope of the existing solar collectors in the hotel.

Figure 4 shows on the psychrometric chart the conditions on an hourly basis for the summer season consisting of the months of June–September corresponding to Madrid (red dots) and Elche (green triangles). The mean and standard deviation of the outdoor dry-bulb temperature were 23.4 °C and 5.99 °C for Madrid and 24.8 °C and 3.53 °C for Elche. As can be appreciated from the values of solar radiation and dry-bulb temperature, the rigor of the meteorological conditions in the summer season is alike in both cities.

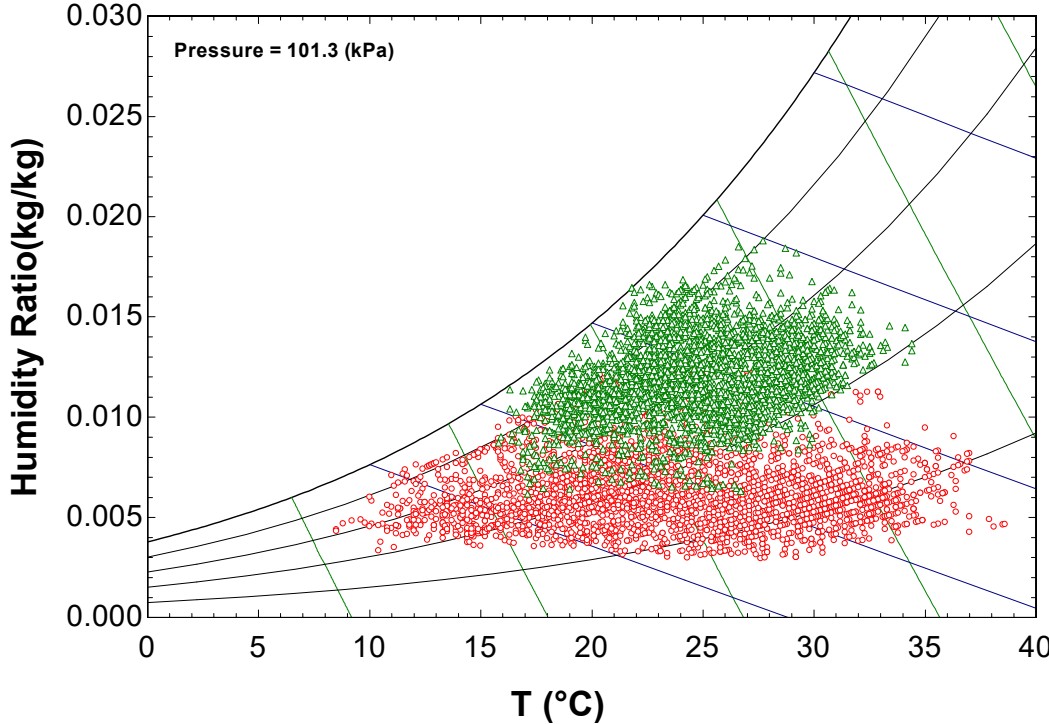

**Figure 4.** Weather conditions on an hourly basis for the summer season (June–September) in Madrid (red dots) and Elche (green triangles).

The thermal energy that is extracted from the hot water storage tank to feed the absorption machine can be translated into cooling capacity by using the EER of the chiller.

There are different ways of modelling the performance of a single-effect absorption chiller using LiBr-H$_2$O as working fluid in the environment of TRNSYS. A first way could be using the Type 107 of the TRNSYS library. However, the manufacturer does not provide operating data for cold water temperatures other than 7 °C for the Yazaki WFC-SC5 installed at the Miguel Hernandez University as well as for the Yazaki WFC-SC10 planned in this work, and therefore Type 107 cannot be used.

A second way could be developing a model based on energy and mass balances and heat transfer equations [8]. This kind of models need the estimation of the heat transfer coefficients and heat exchange areas of the components that constitute the absorption chiller to simulate its behaviour.

The approach implemented in this work was using the EER values obtained by parameterizing the performance curves provided by the manufacturer (Table 3). This approach considers the nominal value of 31 °C for the cooling water inlet temperature and a chilled water outlet temperature of 7 °C.

**Table 3.** EER for driving temperatures of 75, 80 and 85 °C under nominal conditions.

| Tg, in (°C) | Cap (kW) | Qgen (kW) | EER |
|---|---|---|---|
| 75 | 16.6 | 23.0 | 0.722 |
| 80 | 24.3 | 32.5 | 0.748 |
| 85 | 31.0 | 43.0 | 0.721 |

Obviously, the cooling water inlet temperature will vary depending on the outdoor conditions and the cooling tower performance, while the chilled water outlet temperature will be constant if, as it is supposed, the absorption chiller participates in the base cooling demand of the hotel, and whenever it works, it will be at full load.

## 3. Results

Figure 5 represents the average energy delivered per day to the absorption chiller as a function of the specific collector surface along the summer season (June–September) obtained using the TRNSYS model (Figure 1) and the climatic data corresponding to Madrid. As expected, the energy delivered per day increases with the collector field surface.

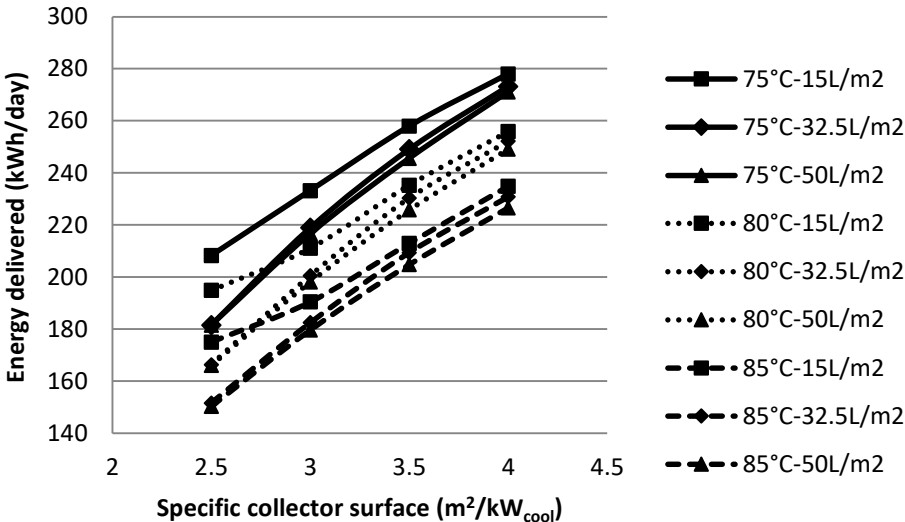

**Figure 5.** Energy delivered per day to the absorption chiller as a function of the specific collector surface.

Figure 5 also shows that the lower the driving temperature (75 °C, continuous line) the higher the energy delivered. This result is logical since the lower the temperature of the fluid that circulates through the solar collectors the greater their efficiency. This trend also agrees with the one registered at the Miguel Hernandez University facility (Table 1).

It can also be seen how the storage volume has a greater effect at low specific collector surfaces. As the collecting surface increases, the effect of the storage volume is weaker. For the driving temperature of 75 °C and a specific surface area of 2.5 $m^2/kW_{cool}$ the difference between the energies delivered corresponding to the specific storage volumes of 15 and 50 $L/m^2$ is 26 kWh/day (208−182), while for the surface area of 4 $m^2/kW_{cool}$ and the same specific storage volumes, the difference is 7 kWh/day (278−271).

The evolution of the temperature in the storage tank for the driving temperature of 75 °C is represented in Figure 6 to study the effect of the specific collector surface and storage volume on the energy delivered to the absorption chiller.

The hot water temperature and energy delivered to the absorption chiller are represented in Figure 6a for the specific collector surface of 2.5 $m^2/kW$ and storage volumes of 15 and 50 $L/m^2$. In Figure 6b the same variables are represented for the specific storage volume of 15 $L/m^2$ and collector surfaces of 2.5 and 4 $m^2/kW$.

The two days represented in Figure 6 correspond to 8 and 9 July (4512 to 4560 ho of the year). Figure 6a explains the difference between the left ends of the curves corresponding to 75 °C in Figure 5. In both cases when the hot water temperature reaches 77.5 °C, 50 kW are withdrawn from the storage tank. This extraction of energy is shorter in the case of the specific storage volume of 50 $L/m^2$ (continuous line) since it takes longer to bring the 4375 L to 77.5 °C with the energy provided by the 87.5 $m^2$ of solar collectors.

The extraction of energy when the temperature of the tank reaches the set-point (77.5 °C), implies a decrease in the temperature of the tank with this collector surface until producing a stop and a second withdrawal in the case of the specific storage volume of 15 $L/m^2$ (discontinuous line). Figure 6b shows that for the specific storage volume of 15 $L/m^2$, the collector surface of 4 $m^2/kW$ (140 $m^2$) is sufficient

to achieve an increase in the temperature of the water in the storage tank simultaneously with the removal of energy for driving the absorption machine. This combination yields the highest energy delivered per day.

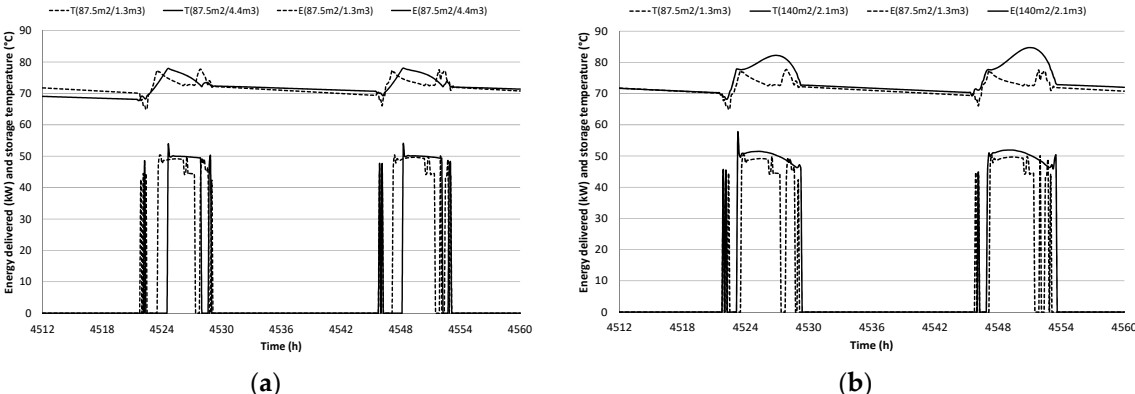

**Figure 6.** Effect of the specific collector field surface and storage volume on the energy supplied to the absorption chiller (**a**) 87.5 m$^2$-1313 L and 87.5 m$^2$-4375 L (**b**) 87.5 m$^2$-1313 L and 140 m$^2$-2100 L. (Driving temperature of 75 °C).

It was somewhat expected that the storage volume more favorable was the minimum analyzed since there is temporary coincidence between cooling demand and solar radiation supply. In view of the results shown in Figure 5, it is worth asking whether it is convenient to reduce the accumulation volume below the 15 L/m$^2$. Lower storage volumes were analyzed and represented in Figure 7.

This figure shows a transition zone between 10–15 (depending on the driving temperature) and 30 L/m$^2$. The increase in accumulation above 30 L/m$^2$ has a negligible effect on the range studied. An additional reduction below 10–15 L/m$^2$ also does not seem to have a positive effect and generates transients that should be analyzed from an experimental point of view. Theoretically, the direct coupling idea (no storage) was evaluated in [18].

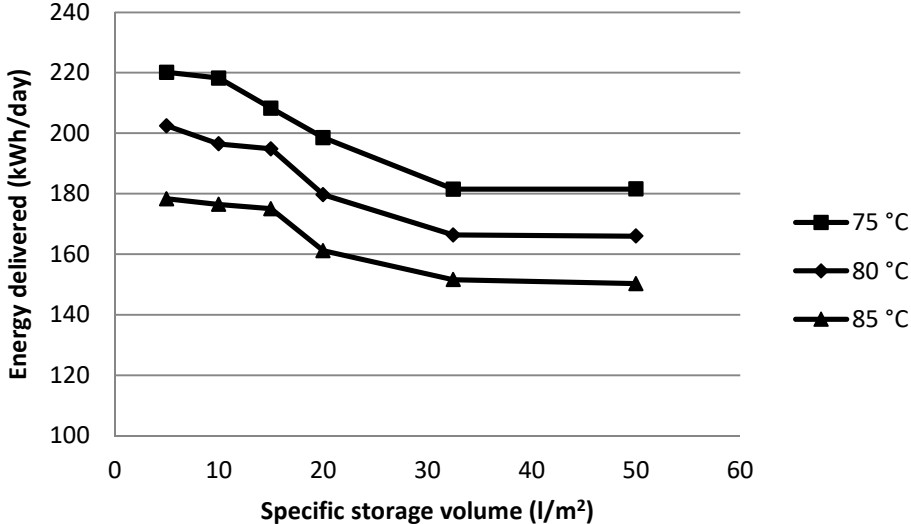

**Figure 7.** Effect of the driving temperature and the specific storage volume on the fraction of solar radiation usable to drive the absorption machine.

Therefore, in terms of the energy delivered to the absorption chiller the optimum dimensioning corresponds to the values of 75 °C for the driving temperature and 15 L/m$^2$ for the specific storage volume.

## 4. Discussion

The cooling capacity calculated from the thermal energy supplied to the absorption chiller and its EER can serve as a basis for an economic study by evaluating its cost when generated by an electric vapor compression chiller.

In the case of a solar collector surface of 122.5 m$^2$ (49 solar collectors), a driving temperature of 75 °C and a specific storage volume of 15 L/m$^2$, 31,466 kWh of thermal energy are sent to the absorption chiller from the hot water storage tank during the summer season (June–September). Considering an EER of 0.722 corresponding to the 75 °C driving temperature (Table 2), the absorption chiller would provide a cooling capacity of 22,718 kWh.

If this cooling capacity were produced by an electric compression chiller with an EER of 3, and considering an electricity price of 0.2 euros per kWh, the absorption chiller would suppose a saving of 1515 euros (per year if it only operates in the summer season). This saving should compensate for the difference between the higher cost of the solar cooling system compared to that of a vapor compression chiller.

However, given the cost of the solar air conditioning system, the economic study is meaningless. The selling price of the absorption chiller (Yazaki WFC-SC10) in Spain (on 29 April 2019) was of 39,620 euros (without taxes). There is only one company that imports this brand of absorption machines in Spain. The cost of the absorption chiller represents only a fraction between 30 and 35% of the solar cooling system total cost (solar collector field, storage tank, cooling tower, water treatment system, etc.) according to the experience of the authors [10]. This means that the total cost of the 35 kW solar cooling facility would be approximately of 105,000 euros (3000 euros per kW of cooling capacity).

The solar cooling system total cost of 3000 euros per kW of cooling capacity is the approximate cost of the facility that the authors designed and built at the Miguel Hernández University of Elche. This cost coincides with that provided by Infante Ferreira and Kim [4] of 2675 euros per kW of cooling capacity. In the case of the Miguel Hernández University facility, the cost of the 17.6 kW cooling capacity Yazaki WFC-SC5 absorption machine was of 24,391 euros (without taxes and with a special discount). At this facility, an open cooling tower was chosen because it was three times cheaper than a closed one. From the point of view of the absorption chiller maintenance it is much more convenient to use a closed cooling tower, even more so considering that the absorption chiller represents a high percentage of the total cost of the solar cooling system.

The economic analysis could be even worse if we took into account the parasitic electrical consumption of the solar energy cooling system. According to the data recorded in the Miguel Hernandez University facility, the cooling capacity of the absorption machine was just 1.7 times higher than the electrical consumption associated with pumps, cooling tower fan and control systems.

The above mentioned price for the absorption chiller includes only one year guarantee. The company that imports the absorption chiller is also the only company that carries out its maintenance in Spain. The maintenance and the initial cost make this system incapable of competing with other air conditioning systems, being therefore limited to demonstration facilities.

Sarabia Escriva et al. [18] describe a method to estimate the upper bound of the cooling energy yield by a solar-cooling facility. It can be applied by hand to get a reference value. Although the details can be found there, here we show its comparison to our practical results. The method assumes that the absorption machine is directly coupled to the collector field and takes into account the thermodynamic performance of both: the solar collectors and the absorption cooling machine. The tables used by the method must be prepared in advance, based on those parameters. In concrete:

- The collector field performance is defined by: orientation (south), tilt angle (15° less than the site latitude), collectors area $A_{col}$ the optical factor $FR_{\tau\alpha}$ and the thermal losses $FRU_L$ (0.825 and 1.1 W/m$^2$K, respectively).
- The absorption machine efficiency is determined using the Felix Ziegler characteristic equation [19]. It has three parameters: thermal size s (kW/K) (i.e., machine size), the heat exchange capacity

allocation $\alpha$ and the internal losses $\Delta T_{min}$. As discussed in [18], a good design will approach $\alpha = 0.5$ and $\Delta T_{min} \approx 0$. As practical values we assumed: $\Delta T_{min} = 3.5$ K and $\alpha = 0.2$, $\alpha = 0.5$ for a bad and good design respectively.

Regarding the operational parameters the method fixes chilled water temperature $T_e$ (5 °C or 14 °C). The generator temperature $T_g$ is variable as the result of an energy balance between the machine and the solar field. The machine is allowed to produce cooling only if $T_g > 75$ °C. The absorber temperature $T_a$ equals the wet-bulb temperature and the condenser one is $T_c = T_a + 2.5$ (°C). Finally, the $\Psi = A_{col}/s$ parameter represents how big is the collector field with respect to the machine size. In our case, for a 35 kW cooling capacity machine its "size" would be s ≈ 1.2 kW/K, thus for $A_{col} = 122.5$ m$^2$, $\Psi = 100$ m$^2$K/kW. The relative size of the driving solar power to the cooling machine is very important and determines the overall cooling driving potential DDGH (K·h) (alike to the degree days concept). The annual cooling effect is computed as:

$$Q_e \text{ (kWh)} = s \cdot DDGH \tag{2}$$

This value is tabulated for each site and for all the aforementioned parameters fixed. For instance, for Madrid producing chilled water at 5 °C and with a good machine $\alpha = 0.5$ table 13 in [18] gives DDGH = 23022 K·h. Therefore the estimated upper bound is $Q_e = 27,626$ kWh while our results provide $Q_e = 22,718$ kWh. A machine designed badly, in the same conditions, would yield $Q_e = 14,072$ kWh. Moreover, the overall performance computed as the ratio between the cooling energy effect and solar radiation impinging onto the collector field in our case is 0.25. Roughly, for Spanish climates, this ratio has a peak value of 0.33 achieved at around $\Psi = 200$ m$^2$K/kW. Our conclusion is that our design is close to be optimal.

## 5. Conclusions

The objective of this work was the design of the components of a solar air conditioning facility for a hotel located in Madrid. The hotel owner wanted to take advantage of an existing field of flat-plate solar collectors to carry out a demonstration project of solar cooling.

A simple model was developed in the TRNSYS program to evaluate the energy delivered from the hot water storage tank to the absorption chiller for the production of cold water, as a function of the collector field surface, volume of the hot water storage tank and driving temperature of the absorption machine.

This model was validated with data registered in the installation and published previously in [17], in terms of the fraction of the solar radiation used to drive the absorption machine. It reproduced the effects of the analyzed variables on the performance of the facility experimentally observed, and provided an ideal slope of the solar collectors for this application of 20°, and optimum values of 75 °C for the driving temperature and 15 L/m$^2$ for the specific storage volume.

The differences between the registered data and the data provided by the model were justified in terms of the use of a typical meteorological year, the difference between the performance of the solar collectors with respect to the efficiency curve supplied by the manufacturer, and thermal losses in pipes.

Nowadays, the main disadvantages of the thermal solar cooling facilities are: the high initial cost of the solar cooling facility, approximately 3000 euros per kW of cooling capacity, the parasitic electrical consumption, the risk of legionella and the water consumption associated to the use of a cooling tower, and the lack of proper maintenance.

The starting conditions of this work (existing field of flat-plate solar collectors) do not affect its conclusions. The solar collector has a high efficiency, although it is true that its slope, as has been proven, is not optimal for a solar cooling application. As for the chiller, the offer of low-capacity absorption chillers driven by hot water is limited.

In the field of solar cooling with absorption machines driven by hot water it is important to work in the direction of reducing the initial cost of the facility, improving the maintenance and eliminating the need of the cooling tower. However, these needs are the same as 20 years ago when the authors began working in this field.

Therefore, more fundamental research is necessary in heat and mass transfer especially in the absorber of the absorption machine to reduce its size. Previous work indicates that it is possible [20]. In the field of maintenance, the study of corrosion (electrochemical analysis) must be deepened, a phenomenon that is produced by the presence of different metals at different temperatures. Air condensation in LiBr-H$_2$O absorption machines is another objective on which research should be focused.

**Author Contributions:** Conceptualization, P.J.M., P.M., V.M.S., L.A.B. and J.R.; methodology, P.J.M., P.M., V.M.S., L.A.B. and J.R.; software, P.J.M. and L.A.B.; validation: P.J.M., P.M.; formal analysis, V.M.S., L.A.B., and J.R.; writing—original draft preparation, P.J.M., P.M.; writing—review and editing: V.M.S., L.A.B., and J.R. All authors have read and agreed to the published version of the manuscript.

**Funding:** This research was funded by (MINECO/AEI/FEDER, UE), grant number ENE2017-83729-C3-1-R.

**Conflicts of Interest:** The authors declare no conflict of interest.

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
