# Peer review of "Design of a 35 kW Solar Cooling Demonstration Facility for a Hotel in Spain"

_applsci, doi:10.3390/app10020496_

Round 1

Reviewer 1 Report

1 while presenting both input and output data, it would be advisable to decrease the presentation of the same in a descriptive manner, applying greater rationality by organizing the values in tables.

2 the approximations that are made in the application of the numerical method and with greater clarity must be described in detail

Reviewer 2 Report

The study is interested and well conducted. The conclusions are already known in the sector, although the used approach can be interesting. Here below some points:

#194 models validation is usually done with same weather condition file in order to avoid differences due to different boundary conditions. If it is not possible to re-run simulations, at least show a weather analysis with the two climates.

#232 The validation of the TRNSYS model is weak as the collector slope is different from the two cases and this is shown also by Figure 4 (difference between 30° and 60°C). Explain why the comparison can be valid despite the different collectors slope.

#279. When considering the storage volume, it is not taken into account the back-up contribution for covering the same demand.

#306 Detail how the costs are calculated and what they refer to.

Last sentence is too generic. Try to focus better on the possible developments on this topic

Reviewer 3 Report

In this paper, authors proposed a study and an experimental validation of a simplified TRNSYS model of a thermal solar cooling system based on absorption chiller. The “tool” was developed by the authors to provide the owner of a facility the information needed for design and decision‐making. To this aim, the authors investigated different collector field surfaces, hot water storage tank volumes and absorption machine driving temperatures. Authors also provided a very trivial economic evaluation.

The treated topic is trendy and up to date, however the approach to the problem is extremely simplistic and the outcomes are neither new or surprising: more complex and accurate models are available for free on the web and in literature since several years. Authors do not provide a comprehensive literature analysis of the field, neglecting important contributions already available. Furthermore, they do not highlight the strengths and the novelty of the work presented (if any).

In general, even though the paper is well written, clear and understandable, it is not worth to be published since it dramatically lacks of novelty and does not contribute to the research in the field of solar cooling.

Round 2

Reviewer 3 Report

In the revised version authors improved the overall merit of the paper, by stating since the beginning the scope of the study, highlighting the simplified approach and focusing more on the target readers. However the work sounds as a project report or a master thesis study. Even though the paper is extremely trivial and un-new it can be a good starting point for further studies in the field or as decision-helper for novice designers and students.
That being said, since authors spent visible efforts for improving the paper and its resulting quality is now sufficient, the paper can be published in its present form.